# AI-Guided Dual Strategy for Peptide Inhibitor Design Targeting Structural Polymorphs of α-Synuclein Fibrils

**DOI:** 10.3390/cells14231921

**Published:** 2025-12-03

**Authors:** Jinfang Duan, Haoyu Zhang, Chuanqi Sun

**Affiliations:** 1Department of Neurology, David Geffen School of Medicine, University of California, Los Angeles (UCLA), Los Angeles, CA 90095, USA; jinfangduan@gmail.com (J.D.);; 2School of Chinese Materia Medica, Tianjin University of Traditional Chinese Medicine, Tianjin 300193, China; 3Departments of Biological Chemistry and Chemistry and Biochemistry, UCLA-DOE Institute, University of California, Los Angeles (UCLA), Los Angeles, CA 90095, USA

**Keywords:** α-synuclein fibrils, peptide inhibitor design, structural polymorphism, artificial intelligence, cryo-electron microscopy, Parkinson’s disease

## Abstract

One of the most important events in the pathogenesis of Parkinson’s disease and related disorders is the formation of abnormal fibrils via the aggregation of α-synuclein (α-syn) with β-sheet-rich organization. The use of Cryo-EM has uncovered different polymorphs of the fibrils, each having unique structural interfaces, which has made the design of inhibitors even more challenging. Here, a structure-guided framework incorporating AI-assisted peptide generation was set up with the objective of targeting the conserved β-sheet motifs that are present in various forms of α-syn fibrils. The ProteinMPNN, then, AlphaFold-Multimer, and PepMLM were employed to create short peptides that would interfere with the growth of the fibrils. The two selected candidates, T1 and S1, showed a significant inhibition of α-syn fibrillation, as measured by a decrease in the ThT fluorescence and the generation of either amorphous or fragmented aggregates. The inhibitory potency of the peptides was in line with the predicted interface energies. This research work illustrates that the integration of cryo-EM structural knowledge with the computational design method leads to the quick discovery of the wide-spectrum peptide inhibitors, which is a good strategy for the precision treatment of neurodegenerative diseases.

## 1. Introduction

Parkinson’s disease (PD) stands as the second largest among the neurodegenerative disorders that affect people all over the world. It is mainly characterized by the gradual deterioration of motor function, the death of dopaminergic neurons in the brain region called substantia nigra, and the formation of abnormal intracellular aggregations known as Lewy bodies and Lewy neurites [1,2,3]. The major protein that forms these inclusions is α-synuclein (α-syn), a 140-amino-acid presynaptic protein that switches from a dissolved, naturally disordered monomer to insoluble β-sheet-rich amyloid fibrils [4,5,6,7]. Such pathological conversion is a characteristic of PD and related synucleinopathies like MSA and DLB [6,8,9].

Cryo-electron microscopy (cryo-EM) has recently made significant progress in revealing that not all α-syn fibrils are of the same kind but rather exist as a whole range of structural polymorphs, each characterized by specific protofilament arrangements, interfacial salt-bridge networks, and β-sheet stacking patterns [10,11]. The different types of this polymorphic fibril have been recognized in biopsies from patients, amplified by seeding, and in recombinant assemblies, showing both disease-specific and common structural features [12,13,14]. Moreover, the different types of fibrils have been found to be biologically active in different ways, hence causing differences in seeding potency, cytotoxicity, and ultimately, disease phenotype [15,16]. One could rightly say that this heterogeneity in structure is the main reason behind the different clinical presentations seen in synucleinopathies.

The various forms of α-syn fibrils can be considered as a major hurdle in the way of the development of new treatments [17]. A lot of small molecules and antibodies deal with the problem by specifically recognizing the different conformations or the corresponding regions of the molecule and hence being unable to stop the action of the other strains of fibrils [18,19]. In addition, it is true that the majority of the currently available inhibitors work only at the very beginning stages of oligomerization and therefore the whole process of fibril elongation remains more or less unhampered [20]. Researchers have found that peptide-based inhibitors can bring about an enormous change and have become the most sought-after in the drug industry owing to their ability to provide very high structural specificity, chemical tunability, and mimicking of natural protein interfaces [21,22]. There is a possibility for short peptides to be designed in such a way as to bind with the aggregation-prone β-sheet motifs, cap the fibril ends, or even destabilize the interactions between the protofilaments. But the enormous variety of possible sequences of peptides that can be combined together and the structural diversity of fibrils make it very hard to find the right peptide sequences that are both effective and have broad inhibitory activity.

Over the past few years, artificial intelligence (AI) has completely changed the field of protein and peptide design [23,24,25]. Among them, the strongest algorithms for structure prediction, like AlphaFold, as well as generative sequence models that have been trained with very large peptide datasets, nowadays allow for rapid scanning of huge sequence spaces with accuracy down to the atomic level [22,26]. The same techniques can be employed to design the inhibitors that can bind to the specified aggregation motifs, and at the same time, be the most stable and soluble ones through physicochemical property optimization [25]. The ability to work with high-resolution cryo-EM templates along with AI revolutionized design and made it a powerful and efficient route for the rational development of peptide inhibitors targeting amyloid fibril polymorphs [27,28,29].

Recent reviews have highlighted that multiple deep learning-based strategies have been developed to rationally target α-synuclein aggregation [30]. As summarized by Allen et al., these approaches generally fall into several conceptual categories: (i) Sequence-level or motif-focused models, which analyze the aggregation-prone NAC region (residues ~61–95) and the acidic C-terminal tail (residues ~96–140) to design peptides that competitively bind or disrupt key amyloidogenic segments. (ii) Structure-guided deep learning frameworks, which use cryo-EM fibril templates to develop peptide or small-molecule binders that target β-sheet grooves, fibril ends, or protofilament interfaces. (iii) Generative machine learning approaches, including peptide language models and reinforcement-learning systems, which create soluble, non-aggregating variants of NAC-like sequences capable of modulating fibrillation. (iv) Models aimed at secondary nucleation surfaces or monomer–fibril interfaces, where deep learning is applied to identify inhibitors that suppress the amplification steps driving pathological propagation. Incorporating this conceptual overview helps contextualize our AI-assisted, dual-strategy design framework within the broader landscape of computational approaches targeting α-syn fibril formation.

In this study, a dual AI-guided design framework was created for the generation of peptide inhibitors that target α-syn fibrils. Initially, the research group mapped out the conserved β-sheet motifs over various cryo-EM-resolved polymorphs and flagged them as the universal inhibitory targets [1,3,5,8,11,31,32,33,34,35,36,37,38,39,40,41,42,43,44]. Computational methods were then applied in two ways: Structure-based design: In this case, ProteinMPNN made the selection of peptides fitting into the grooves and edges of the β-sheet of the fibrils, and the modeling of AlphaFold-Multimer was used for refining the results. Sequence-based design: This involved a peptide language model (PepMLM) that generated the variants coming from the conserved amyloidogenic segments [26,45].

After screening numerous generated candidates, the sequences that were T1–T4 and S1–S4, eight of the highest-ranking ones, were synthesized and put to the test. The synergistic use of Thioflavin-T (ThT) fluorescence assays and transmission electron microscopy (TEM) unveils not only a few peptides but also the two mentioned, T1 and S1, that had very strong dimerizing effects, signaling the decline of α-syn aggregation and modifying the morphology of fibrils. These peptides were getting rid of fibrils by promoting the formation of amorphous or fragmented aggregates rather than mature amyloids.

In summary, this paper presents a methodology that can be applied in a similar way for AI-assisted peptide design. The combination of cryo-EM structural knowledge with that of deep learning-based sequence generation not only enables the easy rational thing to do for discovering the inhibitors of the conserved aggregation motifs but also the process becomes a two-way street. This technique, in conjunction with α-synuclein, can easily disseminate across the board of all amyloidogenic proteins such as tau, Aβ, and TDP-43, thus signifying a bright future in the field of drug development for the treatment of neurodegenerative diseases, via precision therapeutics.

## 2. Materials and Methods

### 2.1. Expression and Purification of Recombinant α-Synuclein

Recombinant human wild-type α-synuclein was expressed in *E. coli* BL21(DE3) cells (New England Biolabs, Ipswich, MA, USA) using a pET28a vector (Novagen, Merck Millipore, Darmstadt, Germany). Bacteria were cultured in Luria–Bertani (LB) medium (BD Difco, Franklin Lakes, NJ, USA) at 37 °C until the OD600 reached 0.6, followed by induction with 1 mM IPTG (Sigma-Aldrich, St. Louis, MO, USA) for 4 h. Cells were harvested by centrifugation at 4000 rpm for 10 min at 4 °C using a Beckman Coulter Allegra X-15R centrifuge (Beckman Coulter, Brea, CA, USA), resuspended in lysis buffer (20 mM Tris-HCl, pH 7.5, 150 mM NaCl), and lysed by probe sonication using a Qsonica Q700 sonicator (Qsonica, Newtown, CT, USA) with 3 s on/3 s off cycles at 60% amplitude for a total of 10 min.

The lysate was clarified by centrifugation at 18,000× *g* for 30 min at 4 °C using a Beckman Coulter Avanti J-26S XP centrifuge (Beckman Coulter, Brea, CA, USA). Bacterial proteins were heat-denatured at 100 °C for 10 min using a digital dry bath (Thermo Fisher Scientific, Waltham, MA, USA). The supernatant was filtered through a 0.45 µm syringe filter (MilliporeSigma, Burlington, MA, USA) and loaded onto a Q-Sepharose Fast Flow anion-exchange column (Cytiva, Uppsala, Sweden) equilibrated with buffer A (20 mM Tris-HCl, pH 7.4). Bound proteins were eluted with a linear NaCl gradient (0–500 mM). Fractions containing α-synuclein were pooled, dialyzed against 20 mM Tris-HCl (pH 7.4) using SnakeSkin dialysis tubing (Thermo Fisher Scientific, Waltham, MA, USA), and further purified by size-exclusion chromatography using a Superdex 200 Increase 10/300 GL column (Cytiva, Uppsala, Sweden) on an ÄKTA Pure chromatography system (Cytiva, Uppsala, Sweden) with 20 mM Tris and 100 mM NaCl (pH 7.4) as running buffer. Protein purity (>95%) was assessed by SDS–PAGE using Mini-PROTEAN TGX gels (Bio-Rad, Hercules, CA, USA) and electrospray ionization mass spectrometry conducted on a Thermo Scientific Q Exactive mass spectrometer (Thermo Fisher Scientific, Waltham, MA, USA). Protein concentration was measured at 280 nm using a NanoDrop One spectrophotometer (Thermo Fisher Scientific, Waltham, MA, USA) with an extinction coefficient of 5960 M^−1^ cm^−1^.

### 2.2. In Vitro Fibrillization of α-Synuclein

The frozen monomeric α-synuclein was thawed and then subjected to ultracentrifugation (100,000× *g*, 30 min, 4 °C) in order to eliminate any preformed aggregates. The preparation of aggregation reactions (100 µL) suggested the use of 100 μM α-synuclein in 20 mM Tris-HCl (pH 7.4), 150 mM NaCl, and 0.01% NaN_3_, with or without the addition of peptide inhibitors at specified molar ratios (α-syn–peptide = 1:0.5, 1:1, or 1:2). In addition, the optical film was used to seal the 96-well microplate (nonbinding surface, black walls, clear bottom), thus preventing evaporation of the samples. The reactions were continuously shaken (orbital, 600 rpm) at 37 °C for a maximum of 72 h.

In the case of seeded reactions, the preformed α-syn fibrils that were produced under identical conditions (no peptides) were sonicated to obtain short seeds (30 s pulse, 10% amplitude) and then incorporated at the rate of 1% (*w*/*w*) into the monomeric reactions. The fibril formation kinetics were tracked by means of Thioflavin-T fluorescence.

### 2.3. Identification of Conserved Structural Motifs from Cryo-EM Data

High-resolution cryo-EM structures of α-syn fibrils from various sources, including patient-derived, seeded, and recombinant, were retrieved from the Protein Data Bank (PDB). Different polymorphs representing various disease states, namely MSA, PD, and DLB, were aligned through the use of PyMOL 2.5 and UCSF ChimeraX-1.1.1 software in order to reveal the structurally conserved β-sheet cores and the interfaces between the different protofilaments (Figure 1). Four conserved β-strand–rich areas (Sites 1–4) were identified according to the following criteria: (i) being present in more than 80% of known polymorphs; (ii) being accessible on the surface at either the ends of the fibrils or the junctions between the protofilaments; and (iii) having a high density of hydrogen bonds. These motifs that were identified as conserved then became the templates for the AI-guided generation of peptides.

### 2.4. AI-Guided Peptide Design Workflow

A dual design approach consisting of both structure-based and sequence-based modeling was applied, and the methodology is presented in Figure 2. During the structure- and sequence-based design process, each top-ranked peptide (e.g., T1 or S1) was accompanied by a set of closely related sequence variants (T1-1, T1-2, …; S1-1, S1-2, …). These variants are automatically generated by ProteinMPNN or PepMLM as alternative solutions within the local sequence space surrounding the parent peptide. They represent small sequence perturbations—typically conservative substitutions or residue permutations—that maintain similar binding geometry but optimize different aspects of the predicted interface, such as side-chain packing, solubility, or backbone compatibility.

These derivative sequences function as local refinements of the parent binder rather than independent designs. Their purpose is to explore fine-grained sequence variation and ensure that the selected representative peptide (T1, S1, etc.) is the highest-scoring member of its local sequence family. Although these variants were evaluated computationally for stability and interface quality, only the top-performing representative from each family (T1–T4 and S1–S4) was selected for chemical synthesis and experimental validation, while the remaining variants (T1-1, T1-2, …) are reported in the Appendix A for transparency and reproducibility.

#### 2.4.1. Structure-Based Peptide Design

For each conserved inhibitory site (Sites 1–4), a local structural environment consisting of interacting β-strands and a 6–15-residue fibril fragment from representative cryo-EM structures was extracted and used as a fixed-backbone template for computational design. ProteinMPNN was applied to generate approximately 5000 peptide sequences per site, optimizing side-chain packing and backbone complementarity at the protofilament interface. All candidates were subsequently evaluated with AlphaFold-Multimer to assess interface confidence, and sequences were retained only if the predicted complexes satisfied mean interface pLDDT > 70, ipTM > 0.40, interface PAE < 6 Å, pose RMSD < 3 Å across three independent AF2 runs, and fibril backbone RMSD < 2 Å. Shortlisted peptides were further assessed using Rosetta InterfaceAnalyzer, requiring ΔG_bind < −8 kcal/mol, buried surface area (dSASA) > 500 Å^2^, shape complementarity (Sc) > 0.60, packing score > 0.55, ≥4 interfacial hydrogen bonds, and clash score < 10. Additional physicochemical filtering included GRAVY index < 0.5, overall charge between −2 and +2, TANGO aggregation propensity < 5%, and exclusion of sequences containing extended hydrophobic patches or predicted self-assembly tendencies. Four top-ranked structure-based peptides fulfilling all criteria were designated T1–T4 for synthesis and experimental testing.

#### 2.4.2. Sequence-Based Peptide Design

A complementary sequence-based design pipeline was constructed using a peptide language model (PepMLM) seeded with four conserved amyloidogenic segments of α-synuclein corresponding to Sites 1–4 (residues 51–56, 58–64, 69–78, and 83–95). For each site, PepMLM generated ~5000 short peptides (9–11 residues) by sampling the model probability distribution (temperature = 0.8, top-k = 50). Initial filtering removed sequences with poor physicochemical properties, retaining only peptides with net charge between −2 and +2, GRAVY index < 0.5, TANGO aggregation score < 5%, and no extended hydrophobic stretches. Remaining candidates were evaluated using AlphaFold-Multimer by docking each peptide to its corresponding fibril template, and only high-confidence complexes were kept (ipTM ≥ 0.60, interface pLDDT ≥ 70, interface PAE < 6 Å, and pose RMSD < 3 Å across triplicate AF2 runs). From the top ~1% highest-scoring sequences—ranked based on interface geometry, predicted stability, and sequence plausibility—four peptides (S1–S4) were selected for synthesis and in vitro validation.

### 2.5. Peptide Synthesis and Characterization

All candidate peptides were synthesized by solid-phase Fmoc chemistry (GenScript, Piscataway, NJ, USA) and purified to ≥ 95% purity by reverse-phase HPLC (C18 column). Molecular weights were confirmed by MALDI-TOF mass spectrometry. Lyophilized peptides were stored at −20 °C until use. Before experiments, peptides were dissolved in ultrapure water or minimal DMSO (<0.1%) and diluted in buffer to final concentrations ranging from 0–140 µM. Concentrations were confirmed by amino acid analysis.

### 2.6. Thioflavin-T Fluorescence Assay

Amyloid formation was monitored by Thioflavin-T (ThT) fluorescence. Reaction mixtures (100 µL) contained 70 µM α-synuclein, 25 µM ThT, and peptide inhibitors at specified concentrations. Fluorescence readings were collected every 10 min using a plate reader (excitation = 440 nm; emission = 485 nm) with intermittent shaking. Increasing ThT fluorescence indicated β-sheet formation. Each experiment was performed with three independent biological replicates, each measured in triplicate. Data were presented as mean ± SD and analyzed by one-way ANOVA followed by Tukey’s multiple-comparison test.

### 2.7. Transmission Electron Microscopy (TEM)

To examine fibril morphology, samples collected at the end of aggregation reactions were adsorbed (10 µL) onto carbon-coated copper grids (400-mesh) for 2 min, blotted, rinsed twice with distilled water, and negatively stained with 2% (*w*/*v*) uranyl acetate for 1 min. After air drying, grids were imaged using a JEOL JEM-1400 TEM (Japan Electron Optics Laboratory Co., Ltd., Tokyo, Japan) operating at 120 kV.

Micrographs were captured at 50,000× magnification and analyzed with ImageJ 1.54r. For each condition, at least ten images from independent grids were quantified. Fibrils were classified as long (>500 nm), short (100–500 nm), or amorphous (<100 nm). The percentage of each morphology type was determined from segmented images. Peptide-only controls (without α-synuclein) were included to assess intrinsic ThT fluorescence.

### 2.8. Statistical and Computational Analyses

All numerical analyses were performed in Python 3.12 (NumPy 2.1.2, Pandas 2.2.2, SciPy 1.14.1) and GraphPad Prism 9.0. Data were reported as mean ± SD. Statistical comparisons included: Aggregation kinetics: analyzed by one-way ANOVA for lag-phase and plateau differences; Morphological distributions: evaluated using chi-square tests; Computational–experimental correlation: assessed by Pearson’s correlation between predicted ΔG values and observed inhibition percentages. All computational workflows (ProteinMPNN v1.1, PepMLM 2024.12 release, AlphaFold-Multimer v2.3.2, Rosetta InterfaceAnalyzer v3.15) were executed using standardized parameters to ensure reproducibility.

### 2.9. Primary Neuron Assay and Immunocytochemistry

Primary cortical neurons from embryonic day 16–18 C57BL/6 mice were plated on poly-D-lysine–coated plates and maintained in Neurobasal medium with B27 and GlutaMAX. At DIV7, neurons were treated with sonicated α-syn PFFs together with peptide inhibitors (S1, S2, T1, T2) at a 1:1 molar ratio (peptide–PFF). Cultures were maintained for 14 days before fixation. For immunocytochemistry, neurons were washed with DPBS and fixed in 4% PFA containing 1% Triton X-100 for 15 min to remove soluble proteins. After blocking with 3% BSA and 3% FBS for 1 h, cells were incubated overnight at 4 °C with primary antibodies against phospho-α-syn (81A, #825701, Biolegend, San Diego, CA, USA), DAPI (D1306, Invitrogen, Carlsbad, CA, USA), and neurofilament light chain (NFL), followed by fluorophore-conjugated secondary antibodies for 2 h at room temperature. Plates were imaged using an ImageXpress Pico (Molecular Devices, LLC, San Jose, CA, USA) automated imaging system under identical acquisition settings. Quantification of α-syn pathology was performed using CellReporterXpress 2.9, measuring the area fraction of 81A-positive aggregates normalized to the PFF-only control. Data represent mean ± SEM from three independent cultures.

### 2.10. Data Availability

All computational scripts, input files, peptide sequences, scoring tables, and predicted structures are publicly available in Appendix A and at the GitHub repository (https://github.com/chuanqisun22/AlphaSyn-Peptide-Design-Supplementary, accessed on 30 November 2025). All raw ThT data and neuron imaging datasets are available from the corresponding author upon request.

## 3. Results

### 3.1. Structural Diversity of α-Syn Fibrils and Identification of Conserved β-Sheet Motifs

For the purpose of discovering universal inhibitory targets, we investigated cryo-EM structures of α-syn fibrils coming from different sources in a systematic manner, such as patient brain tissues, seeded in vitro fibrils and recombinant assemblies (Figure 1 and Appendix A). Notwithstanding the pronounced differences in the protofilament interfaces and cross-β geometries among the various fibril types connected with multiple system atrophy (MSA), Parkinson’s disease (PD) and juvenile-onset synucleinopathy, regardless of polymorphic variation, every fibril type consistently contained the same hydrophobic core formed by residues 37–98 in the NAC region.

The structural alignment of 20 representative fibril polymorphs exposed the presence of four β-strand clusters (Sites 1–4) characterized by recurrent high hydrogen-bond density and very small atomic deviation (RMSD < 1.2 Å), which is a strong indicator of structural conservation of the fibrils. While Sites 1 and 2 were mainly at the fibril ends, giving them the potential for inhibition through end-capping, Sites 3 and 4 were along the inter-protofilament interfaces, creating “cross-hair” or “Greek key” structural motifs.

The analysis of sequence conservation also pointed to recurring hydrophobic and aromatic residues (V66, V70, A76, F94), which made β-sheet contacts that are crucial for the stability of the fibril. Therefore, these four regions were designated as the design templates for the peptide inhibitor development targeting the conserved β-sheet grooves and interfacial pockets responsible for fibril growth and stabilization.

### 3.2. AI-Guided Dual Design and In Silico Screening of Peptide Candidates

The use of the dual AI-guided pipeline resulted in the generation and screening of over 10,000 peptide candidates (6–15 amino acids long) through a combination of structure-based (ProteinMPNN) and sequence-based (PepMLM) methods. After computational filtering of the candidates based on solubility, interface complementarity, and predicted stability, the eight top-ranked candidates T1–T4 (structure-based) and S1–S4 (sequence-based) were selected for experimental validation (Appendix A). Predicted binding models indicated different inhibitory mechanisms for the different candidates (Figure 2). T1 and T2 peptides had their binding longitudinally along β-sheet edges, forming parallel backbone hydrogen bonds that could sterically hinder the addition of the monomer. T3 and T4 interacted perpendicularly across the protofilament interfaces, which may lead to the destabilization of lateral packing.

To aid interpretation of our selection criteria, we note that the scoring metrics reported in Appendix A fall within ranges that are commonly associated with favorable peptide–amyloid or peptide–protein interactions. For ProteinMPNN, more negative per-residue scores typically indicate sequences with stronger geometric fit and lower design energy, consistent with values observed in experimentally validated peptide binders. Rosetta binding free energies (ΔG) in the range of −20 to −35 kcal/mol qualitatively correspond to moderate-to-strong affinity interfaces in systems where Kd values have been measured. For PepMLM, lower pseudo-perplexity values reflect higher sequence plausibility and compatibility with peptide structural statistics, which correlates with improved solubility and reduced aggregation propensity. Although these metrics are not direct determinations of Kd, they provide qualitative indicators of stronger predicted binding and were therefore used to prioritize candidates for synthesis and testing.

S1 and S2 were mimetic of short NAC fragments, but they had hydrophilic substitutions at solvent-exposed positions, thus improving solubility while still being compatible with the β-sheet. S3 and S4 were constructed to develop amphipathic helices that would be able to cap fibril ends through hydrophobic insertion.

Energetic scoring with Rosetta InterfaceAnalyzer gave binding free energies (ΔG) of between −24 and −36 kcal/mol for the leading candidates, while AlphaFold-Multimer predicted interface confidence scores (pTM > 0.6) for T1, T2, and S1 complexes. Such results gave a strong computational backing for the assumption of their high binding affinity and structural compatibility, which in turn led to informed prioritization of subsequent experimental work.

For clarity, sequence variants such as T1-1/T1-2 or S1-1/S1-2 represent local sequence refinements automatically generated by ProteinMPNN or PepMLM during the design process. These variants retain similar predicted binding poses but differ slightly in side-chain packing or solubility properties. Only the highest-scoring representatives (T1–T4 and S1–S4) were synthesized, while other variants are reported in the Appendix A for completeness.

### 3.3. Inhibition of α-Syn Aggregation Monitored by Thioflavin-T Fluorescence

For testing the effect of inhibition, Thioflavin-T (ThT) fluorescence kinetics were taken as a measure for α-synuclein aggregation in the presence or absence of each of the designed peptides (Figure 3). In the control experiments that only consisted of α-synuclein monomers, the typical sigmoidal aggregation curves showed up, which were marked by a lag phase (~16 h), followed by an exponential rise, and then a fluorescence plateau after ~60 h.

On the other hand, all the reactions treated with the peptides showed that the aggregation onset was delayed and the final ThT intensity was diminished, indicating the effective inhibition of β-sheet formation. Of the peptides that were tested, T1 and S1 had the most powerful inhibitory action, almost completely eliminating fluorescence at a 1:2 α-syn–peptide molar ratio. Peptides T2 and S2 provided moderate inhibition, while T3–T4 and S3–S4 resulted in minor suppression.

The decline in fluorescence that was dose-dependent confirmed that the inhibition was connected with the concentration of peptide. Thus, it can be interpreted that the designed peptides are able to interfere with α-syn fibrillation in a concentration-dependent manner, and this, in turn, leads to successful disruption of amyloid propagation. Peptide-only ThT controls showed no intrinsic fluorescence or self-aggregation for any peptide, confirming that inhibition was not due to ThT interference (Appendix A).

### 3.4. Morphological Modulation by Peptide Inhibitors

Negatively stained transmission electron microscopy (TEM) was used as a tool to visualize fibril morphology, and it was done at the end of the aggregation reactions (Figure 4). The untreated control formed a dense network of long, twisted fibrils which averaged about ~800 ± 150 nm in length and ~10 ± 2 nm in diameter.

On the other hand, the samples that were treated with T1 or S1 had almost no mature fibrils present, just some amorphous or short fragmented aggregates (<100 nm) left behind. T2 and S2 treatments caused a significant reduction in the number of fibrils, with mainly short fragments (200–400 nm) and a lower overall density of fibrils. The treatments of T3–T4 and S3–S4 resulted in the formation of mixed populations of short fibrils and disordered aggregates.

The quantitative image analysis verified that there was a reduction of approximately 90% of the long fibrils in the T1/S1 groups when compared to the control (*p* < 0.001). Additionally, the fibril density had a strong positive correlation with the ThT fluorescence intensity (Pearson’s r = 0.91), which indicates the consistency between the fluorescence and TEM results. The combined findings suggested that the peptides, and especially T1 and S1, not only hindered fibril nucleation but also diverted aggregation pathways towards the non-fibrillar assemblies.

### 3.5. Peptide Inhibitors Reduce α-Synuclein Pathology in Primary Neurons

To evaluate the biological relevance of the top peptide candidates, we tested T1, T2, S1, and S2 in primary mouse neurons exposed to α-syn PFFs. Immunostaining with the phosphorylation-specific α-syn antibody 81A showed robust intracellular α-syn accumulation in PFF-only controls, whereas co-treatment with T1 or S1 markedly reduced α-syn pathology, with T2 and S2 showing moderate inhibition. Quantification confirmed significant reductions in 81A-positive signal in T1- and S1-treated cultures (Appendix A). These results support that the most potent inhibitors identified biochemically also suppress α-syn aggregation in a neuronal environment.

### 3.6. Correlation Between Computational Predictions and Experimental Inhibition

The predictive accuracy of the AI-guided design was evaluated by comparing computational parameters—Rosetta binding free energy (ΔG) and AlphaFold interface confidence (pTM)—with experimental inhibition efficiencies. A high degree of correlation was found (r = 0.87, *p* < 0.01), and T1 and S1 were the compounds with the most favorable predicted binding metrics as well as the highest experimental inhibition, indicating a good correlation between the predictions and the experimental results.

The AI-assisted design pipeline was confirmed as predictive and reliable, indicating that structural modeling combined with machine learning-driven sequence generation can successfully prioritize biologically active inhibitors before their synthesis.

## 4. Discussion

The present study establishes a framework for the rational design of peptide inhibitors against α-synuclein (α-syn) fibril polymorphs that is based on structure and informed methods. We discovered small peptides that greatly inhibit α-syn aggregation in vitro by integrating cryo-electron microscopy (cryo-EM)-derived structure data with computational sequence generation. Notably, although the different conformational shapes of α-syn fibril polymorphs are very different from each other, they all have one common feature, i.e., the β-sheet motifs that are conserved and stabilized by interactions of hydrophobic and backbone hydrogen bond types. Such features that occur frequently allow for the creation of constitutively conserved binding sites across different fibril strains that can be utilized for broad inhibition.

The experimental results suggest that the peptides designed have two major pathways through which they affect α-syn aggregation. For instance, T1 and S1 peptides are seen to attach themselves to the ends of the fibrils, thus capping the β-sheet edges and blocking the addition of new monomers. Others like T2 and S2 are presumed to bind at the inter-protofilament contact points; thereby, diminishing the contacts and facilitating the separation of the protofilament interface. The reduction in Thioflavin-T fluorescence and the vanishing of the older fibrils in the transmission electron microscopy images support these mechanisms. The different but concurrent mechanisms imply that the use of conserved β-sheet regions as a target is a reliable strategy to successfully cut off the propagation of the fibrils regardless of their polymorphic variation. The addition of primary neuron data further supports the biological relevance of the designed peptides. To place our results in the context of existing work, we compared our top peptide candidates (T1, T2, S1, S2) with sequences listed in aSynPep-DB [46]. None of our lead peptides showed meaningful sequence overlap with previously reported α-syn inhibitors, supporting the novelty of our AI-guided design approach. The database comparison also highlighted that most known inhibitors cluster around short NAC-derived motifs, whereas our sequences were obtained independently through structural interface design and language model generation. In addition, we included EGCG as a reference inhibitor, and the Appendix A provide an initial functional comparison. We now note in the Discussion that broader head-to-head evaluation with database-derived peptides and small-molecule inhibitors will be an important direction for future work.

One of the major advantages of this method was the complementary approach of structure-based and sequence-based design pipelines. The structure-based technique, which was supported by ProteinMPNN and AlphaFold-Multimer, guaranteed geometric and energetic complementarity with atomically detailed templates, while the sequence-based model (PepMLM) allowed for the wider exploration of sequence space beyond the known amyloidogenic motifs. The most potent inhibitors, T1 and S1, were, importantly, obtained through different but coalescent computational techniques, thus emphasizing the blending of orthogonal design paradigms. The excellent agreement between computational predictions and experimental potency is additional proof that binding energy and confidence metrics can be used as trustworthy pre-screening criteria, thus cutting down on the experimental workload and enhancing design efficiency [47,48].

An important consideration for future translational development is improving the stability of short linear peptides, which are naturally prone to rapid proteolysis and limited bioavailability. Several well-established strategies can enhance their in vivo performance, including D-amino acid substitution to increase protease resistance, backbone or head-to-tail cyclization to reduce conformational flexibility, incorporation of N-methylated or non-canonical residues to limit enzymatic degradation, and conjugation to PEG or nanoparticle carriers to prolong circulation and support potential blood–brain barrier transport. These stabilization approaches are fully compatible with our AI-guided design framework and can be readily integrated into future iterations to advance the most promising inhibitors toward in vivo evaluation and therapeutic development.

From the viewpoint of therapy, the aggregation of α-syn is the major process in the development of Parkinson’s disease (PD) and essentially all types of synucleinopathies [20,21,49]. During the current approaches using small molecules and antibodies, the attempts at therapeutic intervention have not been very effective, in part, due to the many different conformations of the α-syn aggregates [31]. The present work on the peptide inhibitors has the possibility to target the structural features that are universally present in all polymorphs, thus being a potential solution for overcoming the issue [21]. The peptide’s unique characteristics of being modular and having adjustable chemistry make them eligible for additional improvements in terms of stability, brain permeability, and resistance to metabolizing by using backbone cyclization, D-amino acid substitution, or nanoparticle conjugation.

This study has several limitations that must be acknowledged. All tests were performed in vitro, which are conditions that do not entirely mimic the intracellular environment. The work planned for the future should check if the peptides work against α-syn aggregation and toxicity in neuronal and in vivo models. It will be necessary to carry out high-resolution structural characterization of peptide fibril complexes using methods like cryo-EM or solid-state NMR to get the binding modes confirmed and the mechanistic understanding refined. Moreover, kinetic analyses separating the nucleation and elongation rate constants could assist in identifying the stage-specific inhibitory effects. Lastly, pharmacokinetic profiling and AI-assisted stability and delivery optimization will be indispensable to getting these peptides ready for therapeutic use.

## 5. Conclusions

This study discloses a combined computational-experimental method for designing peptide inhibitors that will, more or less, target α-syn fibril polymorphs. The integration of cryo-EM structural insights with algorithm-guided sequence design enabled the detection of short peptides that could severely moderate α-syn fibrillation and change fibril morphology in vitro. The best-performing ones, T1 and S1, almost wholly prevented amyloid formation through terminal capping and interfacial disruption mechanisms.

The given evidence points out that the fusion of structural biology with computational peptide design quickens and makes rational the discovery of aggregation inhibitors. The methodology developed at this point is applicable to other amyloidogenic proteins like tau, Aβ, and TDP-43 and lays the groundwork for precision-engineered therapeutics that counteract protein misfolding in neurodegenerative diseases.

## Figures and Tables

**Figure 1 cells-14-01921-f001:**
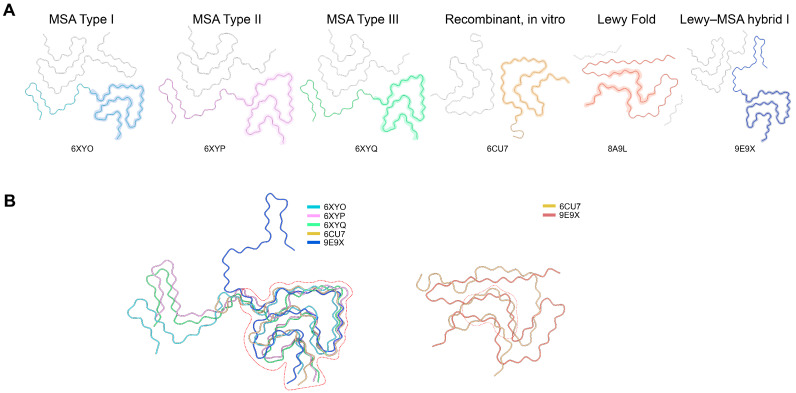
Structural landscape of α-synuclein fibril polymorphs and their conserved core architecture. (**A**). Cryo-EM structures of major α-synuclein fibril polymorphs originating from multiple system atrophy (MSA Type I–III; PDB: 6XYO, 6XYP, 6XYQ), recombinant in vitro assemblies (6CU7), the Lewy fold (8A9L), and the Lewy–MSA hybrid fold (9E9X). Each panel highlights the characteristic protofilament arrangement and topology that distinguish individual fibril types. (**B**). Structural overlays of representative polymorphs reveal a conserved β-sheet–rich core shared across disease-derived and recombinant fibrils (dashed outline), despite pronounced variations in peripheral segments and protofilament packing. This conserved kernel underpins the common structural features of α-syn aggregation and provides a rational basis for identifying universal inhibitory sites.

**Figure 2 cells-14-01921-f002:**
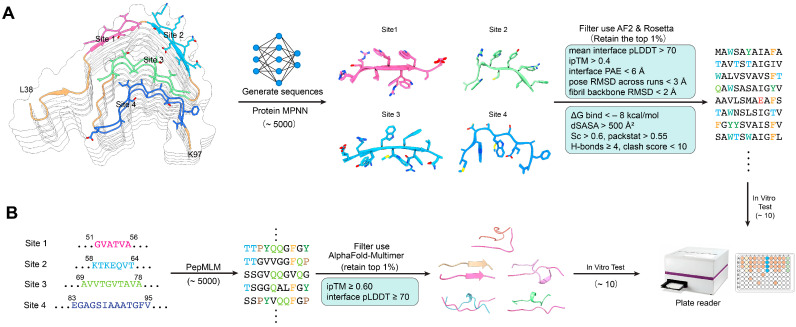
Dual design strategies for peptide inhibitors targeting conserved structural motifs of α-synuclein amyloid fibrils. (**A**). Structure-based design. Conserved binding sites (Sites 1–4) on the α-synuclein fibril architecture were identified as potential capping positions to interfere with fibril elongation. ProteinMPNN was employed to generate diverse candidate peptide sequences, which were subsequently filtered and ranked using AlphaFold2 (AF2) and Rosetta to assess binding compatibility and structural stability. (**B**). Sequence-based design. Conserved fibril segments were used as seed motifs to drive peptide generation via PepMLM, a sequence-based generative language model. The resulting candidates were screened using AlphaFold-Multimer to predict their binding modes with fibril segments. Shortlisted peptides from both design strategies were subjected to in vitro functional assays using a plate reader to evaluate inhibitory efficacy. Colors in the figure indicate the four conserved binding sites on the α-synuclein fibril (Site 1: pink, Site 2: green, Site 3: cyan, Site 4: blue).

**Figure 3 cells-14-01921-f003:**
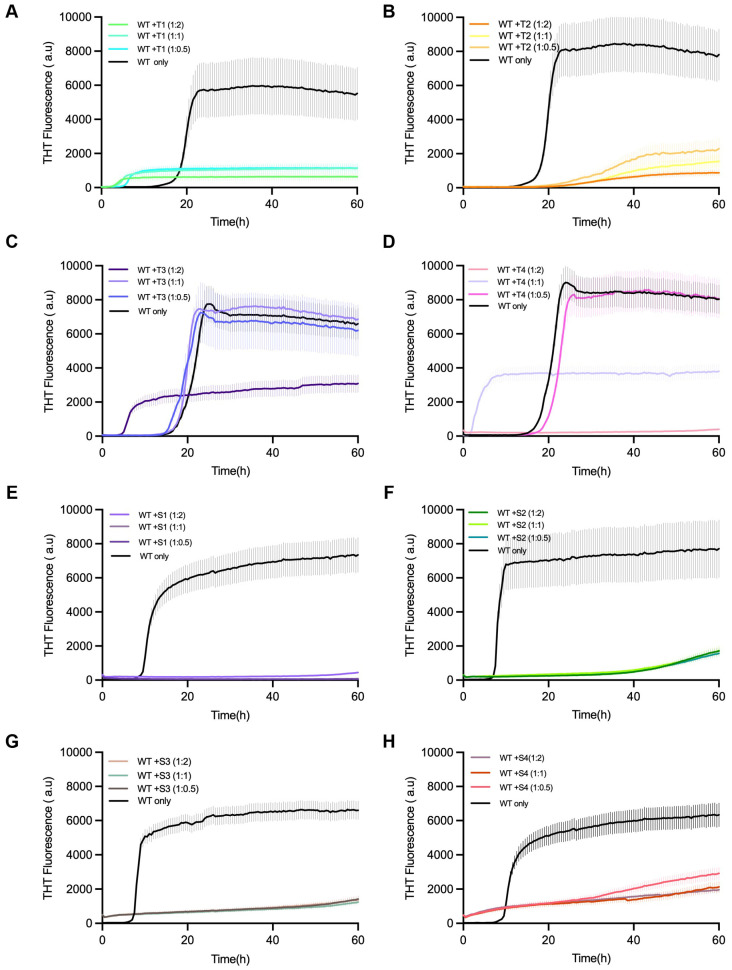
Inhibition of α-synuclein fibril formation by designed linear peptides T1–T4 and S1–S4. Thioflavin T (ThT) fluorescence assays showing the aggregation kinetics of α-synuclein monomers in the presence of designed linear peptide inhibitors. (**A**–**D**) Correspond to T1–T4 peptides; (**E**–**H**) correspond to S1–S4 peptides. α-Synuclein monomers (70 µM) were incubated alone (black) or with each peptide at molar ratios of 1:0.5, 1:1, and 1:2 (monomer–peptide). ThT fluorescence was monitored over 60 h to assess amyloid fibril formation. All peptides exhibited inhibitory effects on α-synuclein aggregation to varying degrees, reflected by reduced fluorescence intensity and prolonged lag phases compared to WT-only controls. Data are presented as mean ± s.d. from three independent replicates.

**Figure 4 cells-14-01921-f004:**
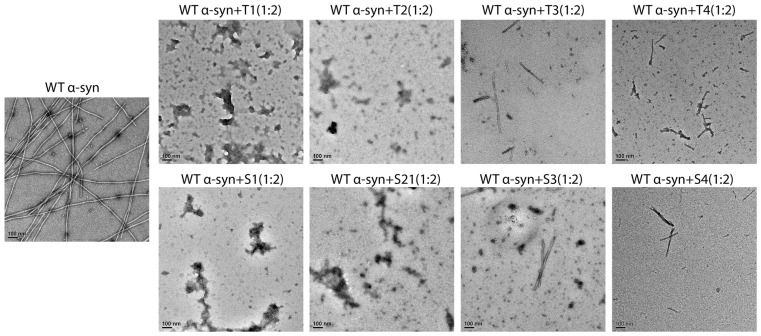
Representative TEM images illustrating the morphological effects of designed peptide inhibitors on α-synuclein fibrillation. Representative negative-stain micrographs illustrate the morphological outcomes of α-syn aggregation in the absence or presence of peptide inhibitors. Untreated WT α-syn formed abundant long, well-defined fibrils. T1 and S1 produced the strongest inhibitory effects, with mature fibrils nearly absent and predominantly amorphous or small granular aggregates observed. T2 and S2 induced partial inhibition, yielding reduced fibril numbers and shorter fragments. T3, T4, S3, and S4 showed comparatively weaker effects, with short fibrils and partially disassembled intermediates remaining. Scale bar: 100 nm.

## Data Availability

All data supporting the findings of this study are available from the corresponding author upon reasonable request. Cryo-EM models analyzed in this work were obtained from the Protein Data Bank, with accession codes listed in Figure 1 and Appendix A. All computational inputs (ProteinMPNN backbone templates, PepMLM seed sequences, AlphaFold-Multimer input files), full scoring tables, and predicted complex structures are provided in Appendix A. The full reproducibility repository, including all computational scripts, input files, and example outputs, is publicly available at: https://github.com/chuanqisun22/AlphaSyn-Peptide-Design-Supplementary (accessed on 30 November 2025).

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
