# Peer review of "AI-Guided Dual Strategy for Peptide Inhibitor Design Targeting Structural Polymorphs of α-Synuclein Fibrils"

_cells, 2025, doi:10.3390/cells14231921_

Round 1
Reviewer 1 Report
Comments and Suggestions for Authors
This study describes a computational–experimental workflow that integrates structure-based modeling (ProteinMPNN, AlphaFold-Multimer) and sequence-based peptide generation (PepMLM) to design short peptide inhibitors targeting conserved β-sheet motifs in α-synuclein fibrils. Among the designed candidates, two peptides (T1 and S1) showed strong inhibition of fibril formation in vitro, supported by Thioflavin-T fluorescence and transmission electron microscopy.
The work addresses a relevant and timely problem in the field of neurodegenerative diseases and provides a clear strategy for rational peptide design informed by structural data. The manuscript is well organized and technically detailed. However, some aspects of the experimental validation and mechanistic interpretation remain preliminary or insufficiently supported by quantitative data. In particular, the biological and therapeutic implications should be presented more cautiously.
Overall, the study offers a promising proof of concept for structure-guided peptide inhibitor development but requires additional validation and contextual discussion to meet publication standards for Cells.
Major Comments
-
Experimental validation is limited to biochemical assays.
The conclusions about inhibitory potency and potential therapeutic value are based only on ThT fluorescence and TEM observations. If possible, additional tests in neuronal or cell-based systems would be needed to demonstrate biological relevance. -
Mechanistic interpretation remains speculative.
The proposed “end-capping” versus “interface-disruption” mechanisms are inferred from modeling but not experimentally confirmed. Kinetic analyses (e.g., separation of nucleation and elongation phases) or seeding experiments would help substantiate these mechanistic hypotheses. -
Potential peptide self-aggregation or ThT interference.
Control data excluding peptide-induced fluorescence or aggregation are not shown. Experiments testing ThT fluorescence of peptides alone would clarify whether the observed inhibition reflects possible interference with α-syn fibrillation. -
Comparison with previously reported inhibitors.
It would strengthen the manuscript to compare the activity of T1 and S1 with known α-syn-targeting peptides such as PQK7 (ref. 21) or other short anti-amyloid sequences.
Minor Comments
-
Figure 2 could include representative quantitative details (e.g., number of peptides generated and filtering thresholds) to better illustrate the design workflow.
-
Please indicate whether the designed peptides were screened for sequence overlap with known α-syn aggregation-prone regions to ensure specificity.
-
In the Discussion, consider briefly outlining strategies to enhance peptide stability (e.g., D-amino acid substitution, cyclization), as these are important for potential translational applications.
-
Typographical note: “ai-assisted” should be consistently written as “AI-assisted.”
Reviewer 2 Report
Comments and Suggestions for Authors
The manuscript by Duan et al. describes the design of peptide inhibitors against alpha-synuclein aggregation. The field is highly competitive and there is room for novel concepts and approaches. The work contains novel results from a machine learning-based pipeline validated by experiments. My main concern is about the presentation of the work, for which I suggest improvement both in terms of placing the work better within this rapidly developing field and clear description of the methodology to enhance reproducibility.
Major points:
- the manuscript cites work on the deep learning-based approaches targeting alpha-synuclein aggregation, but mostly without any specific details. I suggest including a short overview in that the Introduction on the different specific strategies – what regions are targeted and what is the main concept in selected studies, see e.g. the review by Allen et al. 2023, Molecular Neurodegeneration (2023) 18:80 (https://doi.org/10.1186/s13024-023-00675-8)
- Even more importantly, I kindly ask the authors to provide an evaluation of the results in the light of the literature. Especially, comparing the obtained sequences to those listed in aSynPep-DB could be informative (Pintado-Grima et al. 2023, Database 2023, 1-9, https://doi.org/10.1093/database/baad084), along maybe with an evaluation of its accompanying algorithm. In addition, a(t least a short) comparison of the effect obtained with other peptide/small molecule inhibitors would be very welcome.
- Please provide more information on the design process. It would be crucial to see the actual input data and parameters used for AlphaFold-Multimer and PepMLM. For the latter, the sentence “The model performed by placing hydrophilic substitutions on the solvent-exposed sites that required hydrogen-bond compatibility for solubility reasons.” is suggested be completely rephrased to provide a clear description on the input.
- Tables 1 and 2 list lots of sequences, and the manuscript does not seem to offer any clues of the T1-1, T1-2 etc. sequences, their similarity to the (parent?) T1 is nontrivial and I could find no explanation whatsoever. I kindly ask the authors to consider moving these sequences to Supplementary material but providing an informative description anyway.
- I would suggest providing more background data in a freely accessible format (either as direct supplementary material or e.g. on Zenodo). I think that the actual inputs for the AI tolls, the coordinates for the obtained complexes, the different binding strength estimates in numerical format etc. would be useful.
Minor points:
- Lines 248-249: please rephrase “all types still could not avoid the common hydrophobic core of residues 37–98” to clarify what the authors intended to say here.
- When listing peptide sequences (like in Tables 1 and 2), I suggest the use of a fixed-width font (like Courier).
- Please explain Pseudo Perplexity in Table 2.
- Figure 1 could be simplified focusing on the common motifs with the current version moved to Supplementary material.
- In the caption of Figure 4, I advise to indicate that the images shown are representative of many (if possible, please specify, how many) images obtained. The present caption is too long, the experimental procedure does not need to be repeated here, I suggest to include just the most important observations here (inhibition of the aggregation).
Comments on the Quality of English LanguageThe overall English of the manuscript is OK but there are some very strange sentences, I highlighted two of the in my review. The manuscript could benefit from a thorough re-reading and rephrasing the long, complicated sentences.
Reviewer 3 Report
Comments and Suggestions for Authors
The manuscript by Duan et al., titled “AI-Guided Dual Strategy for Peptide Inhibitor Design Targeting Structural Polymorphs of α-Synuclein Fibrils”, presents an interesting study where authors incorporated two different approaches for AI-guided peptide-base inhibitors development for the treatment of Parkinson disease. One approach, utilizing ProteinMPNN algorithm, is structure-based, the other, with PepMLM generator as a starting point, is sequence based. Both have shown promising outcomes with best hits significantly reducing α-synuclein fibrils formation during follow-up in-vitro experiments using Thioflavin-T fluorescence assays. This research illustrated that the integration of cryo-EM structural knowledge with the computational design method can lead to the quick discovery of the wide-spectrum peptide inhibitors in heterogenous systems, potentially applicable to other neurovegetative diseases. I think that data presented is solid and is worth publication in Cells. I have a couple of minor comments:
- The authors posit that data is available upon “reasonable” request, and since there could be variations in what is considered “reasonable” I suggest putting as much data as possible (at least peptide sequences and computational scripts) into the Supplementary materials, so that people could test the algorithms on presented database themselves if interested.
- The description of scores in Tables 1 & 2 could be more explicit - Which numbers are considered to be good and how are they compared to the numbers found in systems with known Kds of interactions?
- With two orthogonal approaches seems to provide comparable in efficacy results (T1 vs S1) for this system, some discussion would be interesting of what the authors think about this outcome in general, would they expect the same for other systems? Have they seen somewhat overlapping sequences in peptides from different approaches showing different ratings in each category (at least this looks like to me)? What could that mean? Are these approaches really orthogonal? Which is better? Do we need to use both in future? These are all speculations at this point, but it would be interesting to know what the authors think.
Round 2
Reviewer 1 Report
Comments and Suggestions for Authors
The study clearly falls within the journal’s scope and presents results that are highly relevant for the rational design of peptide-based inhibitors targeting amyloid fibril polymorphs.
The authors have comprehensively addressed the previous reviewers’ comments by improving the clarity of the manuscript and strengthening the methodological descriptions and related explanations.
Therefore, I recommend the manuscript for acceptance.
Reviewer 2 Report
Comments and Suggestions for Authors
I thank the authors for their efforts to improve the manuscript.